# SLaNC: Static LayerNorm Calibration

**Mahsa Salmani, Nikita Trukhanov, Ilya Soloveychik**
d-Matrix
{msalmani, ntrukhanov, ilyas}@d-matrix.ai

## Abstract

The ever increasing sizes of Large Language Models (LLMs) beyond hundreds of billions of parameters have generated enormous pressure on the manufacturers of dedicated hardware accelerators and made the innovative design of the latter one of the most rapidly expanding fields of the AI industry. Various approaches have been explored to enable efficient and accurate processing of LLMs on the available accelerators given their computational and storage limitations. Among these, various quantization techniques have become the main focus of the community as a means of reducing the compute, communication and storage requirements. Quantization to lower precision formats naturally poses a number of challenges caused by the limited range of the available value representations. When it comes to processing the popular Transformer models on hardware, one of the main issues becomes calculation of the LayerNorm simply because accumulation of the variance requires a much wider dynamic range than the hardware enables. In this article, we address this matter and propose a computationally-efficient scaling technique that can be easily applied to Transformer models during inference. Our method suggests a straightforward way of scaling the LayerNorm inputs based on the static weights of the immediately preceding linear layers. The scaling factors are computed offline, based solely on the linear layer weights, hence no latency or computational overhead is added during inference. Most importantly, our technique ensures that no numerical issues such as overflow or underflow could happen during the compute. This approach offers smooth, accurate and resource-effective inference across a wide range of hardware architectures. The article provides theoretical justification as well as supporting numerical simulations.

## 1 Introduction

Large Language Models (LLMs) based on Transformers [1] have recently become the dominant Deep Neural Network (DNN) architecture due to their unprecedented performance results in all language modeling [2, 3], text processing [4], image and video generation [5], and many other tasks. However, this success comes at a cost of enormous volumes of compute, storage, and data transfer. A whole new industry of dedicated hardware accelerators has emerged in the last few years to accommodate the needs of LLM training and inference [6, 7]. Another major initiative targeted at making the inference feasible and sustainable involves the development of lower precision formats [8, 9, 10], efficient quantization techniques [11], algorithmic solutions [12], accurate approximations [13], and other software optimizations [14, 15].

Efficient quantization techniques such GPTQ [16], AWQ [17], SmoothQuant [18], KVQuant [19], K-sort [20], and numerous others enable storing and processing of LLMs in low-precision formats. Often, that would involve training the model in FP32 format and casting it to 4, 8 or 16-bit precision formats before deployment onto inference hardware [11, 21, 20]. The most popular approach is to compress the static weights to 4 or 8-bit integers or floats and reduce the activations to FP16 or BF16

Submitted to the Second Workshop on Machine Learning with New Compute Paradigms at NeurIPS (MLNCP 2024). Do not distribute.

[22]. In this paper, we focus on the wide family of accelerators operating on FP16 activations for their popularity [23, 24] and specifically for the relatively narrow dynamic range (the range of representable numbers) of FP16 which might pose significant computational challenges. The most critical manifestation of this problem occurs during the LayerNorm computation. Importantly, inclusion of dozens or even hundreds of LayerNorm operators in current Transformers is unavoidable since they prevent the gradients from exploding or decaying during training [25]. At inference, though, processing LayerNorms on accelerators is extremely challenging because they require accumulation of squares of the inputs for the sake of variance (and norm) calculation [26]. Accumulation of such a large number of positive values in FP16 is almost surely bound to overflow.

In this work, we address this problem and propose an efficient, theoretically justified, and easy to implement scaling technique that leads to complete elimination of the FP16 overflow (or underflow) issue in LayerNorms. First, note that scaling of the LayerNorm input does not affect the output due to the homogeneity of the normalization operation but can very significantly shift the range of the accumulated numbers in the denominator computation. Based on this observation, we developed the SLaNC (Static LayerNorm Calibration) method which provides succinct closed formulae for scaling the inputs of all LayerNorms of any Transformer. Importantly, the SLaNC scales are computed solely based on the static weights of the preceding linear layers, and can be therefore computed offline without impacting the inference runtime. The formulae suggested by SLaNC are theoretically justified by derivations and detailed explanations and only involve norms of static weight matrices that can be directly and precisely computed using standard software.

The rest of the article is organized as follows. First, we outline the notation, then in Section 2 we formulate the numerical problem caused by the LayerNorm computation in FP16. Section 3 presents the SLaNC technique together with its theoretical justification. Supporting numerical simulation on the Llama family of LLMs are demonstrated in Section 4. The concluding remarks can be found in Section 5.

**Notation.** The following notation is used in the article. Matrices are denoted by capital bold letters $\mathbf{M}$ and vectors by lower case bold $\mathbf{v}$. The operator product of matrices $\mathbf{A}$ and $\mathbf{B}$ of appropriate sizes is written as $\mathbf{A} \cdot \mathbf{B}$ or $\mathbf{A}\mathbf{B}$, while their element-wise product would be denoted by $\mathbf{A} \odot \mathbf{B}$. For matrix $\mathbf{M}$, we write $\|\mathbf{M}\|_F$ for its Frobenius norm and $\|\mathbf{M}\|$ for its spectral norm; for vector $\mathbf{v}$, by $\|\mathbf{v}\|$ we denote its Euclidean norm. Given vector $\mathbf{m}$, we denote by $\mathbf{M} = \mathrm{diag}(\mathbf{m})$ the diagonal matrix with elements of $\mathbf{m}$ on the main diagonal.

## 2    Problem Formulation

Quantization of an LLM to a low-precision format (e.g., 4, 8 or 16-bit) can lead to a significant degradation of the output quality, and thus has to be applied together with some advanced technique capable of restoring the accuracy [16, 17, 18, 19, 20, 27, 28]. However, an even bigger challenge caused by casting models into low-precision formats is the limited dynamic range of such formats, which can completely ruin the compute flow if applied blindly. The most prominent example is the computation of LayerNorm, which becomes impossible on FP16 accelerators due to the unavoidable overflows and underflows as demonstrated next.

### 2.1    LayerNorm Compute

Layer Normalization (LayerNorm) has become one of the most ubiquitous non-linear operations in modern DNNs since it prevents the gradients from decaying or exploding during training. Extensive literature has demonstrated that the current DNN architectures cannot be practically trained without frequent normalization of hidden states [29, 30, 31]. State of the art Transformer models include dozens or even hundreds of LayerNorm operators which are introduced to facilitate training but make inference troublesome due to the numerical problems introduced by the computation of their denominators.

Given a row input $\mathbf{x} \in \mathbb{R}^d$ and fixed parameters $\boldsymbol{\gamma}, \boldsymbol{\beta} \in \mathbb{R}^d$, the LayerNorm output reads as

$$\mathbf{y}(\mathbf{x}) = \left(\frac{\mathbf{x} - \mu\mathbf{1}}{\sigma}\right) * \boldsymbol{\gamma} + \boldsymbol{\beta} = \left(\frac{\mathbf{x} - \mu\mathbf{1}}{\sigma}\right)\boldsymbol{\Gamma} + \boldsymbol{\beta} \in \mathbb{R}^d, \tag{1}$$

where $\mathbf{1} \in \mathbb{R}^d$ is the vector of ones, $\mathbf{\Gamma} = \mathrm{diag}(\boldsymbol{\gamma})$, and

$$\mu = \frac{1}{d}\sum_{i=1}^{d} x_i, \quad \text{and} \quad \sigma = \sqrt{\frac{1}{d}\sum_{i=1}^{d}(x_i - \mu)^2} = \sqrt{\frac{1}{d}\sum_{i=1}^{d} x_i^2 - \mu^2}. \tag{2}$$

As Eq. 2 suggests, the standard way of computing $\sigma$ requires summing up the squares of the input vector elements. Depending on the range of these elements, such accumulation can easily lead to an overflow or underflow when performed in FP16 or FP8 formats. It is important to note that the majority of the available LLM accelerators process non-linear operations exclusively in FP16 format [32, 33, 34]. While some accelerators do support FP32 accumulation in non-linear modules, this option often comes at a high latency increase making FP32 regime impractical. That is because many SIMD engines can only process FP16 data in the vectorized fashion but not FP32. Fig. 3c and Fig. 3a show the typical distributions of the sum of squares from Eq. 2 in one of the layers of Llama-2. We observe that in too many cases the resulting values exceed the range of FP16, leading to invalid inference.

Note also that the Transformer architecture comes in two flavors based on the location of the residual branch-out. It can take off before the LayerNorm (pre-LN residual) or after (post-LN residual), Fig. 1. Originally, the post-LN option was suggested [1] but later the other one became quite popular since it was observed to speeds-up the training [35]. To be specific and for lack of space below we focus on the post-LN desugn, however, we emphasize that the derivations and conclusions equally apply to the pre-LN one.

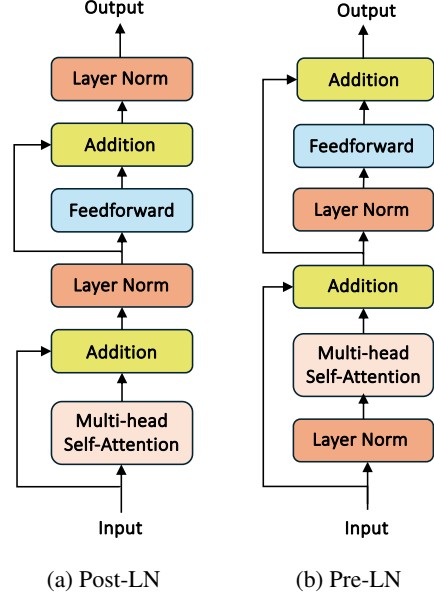

(a) Post-LN    (b) Pre-LN

Figure 1: Residual branching options.

# 3 LayerNorm Scaling

## 3.1 Dynamic Model Calibration

The natural way of addressing the problem of overflow or underflow during computation of Layer-Norm would be to appropriately scale its input. Determining the correct scaling factors appears to be challenging because while avoiding overflow we also do not want to excessively dump the input causing underflow and vice versa. As a consequence, any reasonable scaling algorithm must take into account the actual LayerNorm input values and cannot set the scaling parameters blindly.

A common solution would be to calibrate the scaling factors. This involves passing a test dataset through the Transformer to gauge the range of the input vector norms and setting the scaling factor based on some first-order statistic of this range (e.g., mean or median norm). This technique requires extra calibration data and significant computational overhead even for such a basic operation as LayerNorm, making this approach impractical.

## 3.2 Analytical Static Scaling

In this work, we propose a different methodology that enables analytical offline computation of the desired scaling factors. The scales are determined solely based on the static weights of the linear layers immediately preceding the LayerNorm at hand. This way we calibrate all the LayerNorm operators of a model statically, without using a calibration dataset or additional runtime resources — everything is computed preemptively during model compilation.

The idea of the method is based on a simple observation that LayerNorms inside a Transformer occur frequently and in a regular pattern since any large Transformer is a chain of dozens of identical decoders. Typically, two consecutive LayerNorms surround the attention or the Multi-Layer Percep-tron (MLP) block of every decoder. Eq. 1 suggests that we can treat a LayerNorm as a Euclidean

normalization followed by a diagonal matrix multiplication.[1] From this natural decomposition of the LayerNorm operator we infer that immediately after normalization (the first step in LayerNorm), the norm of the hidden vector $\mathbf{x}$ is equal to one. Our goal is to trace the computational graph from this point to the next LayerNorm and gauge the orders of magnitude of $\|\mathbf{x}\|$ changes based on the transformations it undergoes along the way.

### 3.3 SLaNC for Standard MLP Block

To illustrate the idea, let us consider the MLP block of a standard Transformer, Fig. 2a. Since we neglect the additive bias $\boldsymbol{\beta}$, the output of the MLP block can be expressed as

$$\mathbf{y} = \mathcal{F}(\mathbf{x}\boldsymbol{\Gamma}\mathbf{E})\,\mathbf{G} + \mathbf{x}\boldsymbol{\Gamma}, \tag{3}$$

where the addition comes from the residual connection, and $\mathcal{F}(\cdot)$ is an element-wise non-linearity which is usually a contraction function (e.g. ReLU, GeLU, etc.) making the norm of its argument smaller. Since usually, the maximal partial derivative of $\mathcal{F}(\cdot)$ is bounded by a constant close to one, we can approximate the norm of $\mathbf{y}$ as

$$\|\mathbf{y}\| \propto \|\mathbf{x}\boldsymbol{\Gamma}\mathbf{E}\mathbf{G} + \mathbf{x}\boldsymbol{\Gamma}\|_F. \tag{4}$$

Eventually, we conclude that

$$\frac{\|\mathbf{y}\|}{\|\mathbf{x}\|} \propto \|\boldsymbol{\Gamma}(\mathbf{E}\mathbf{G} + \mathbf{I})\|_F. \tag{5}$$

Recall that $\mathbf{x}$ is the output of the normalization step of a LayerNorm (see Fig. 2a) and thus has unit norm. Therefore, it is natural to set the scaling factor of the following LayerNorm to the right-hand side of Eq. 5 and this should solve the overflow/underflow issue. In Section 4, we demonstrate by extensive simulations that this is actually the case. Note that the scale determined by Eq. 5 only involves static weights and can be computed offline.

### 3.4 SLaNC for Llama MLP Block

Using the same methodology, we derive an analogous formula for the scaling factors of the Layer-Norm following the modified MLP block designed for the decoders of the Llama family of models, Fig. 2b. Here, in addition to the two linear layers of the standard MLP block, we have another linear layer whose output is multiplied with the output of the non-linearity in the element-wise manner. The non-linear function itself is usually chosen to be GeLU. The input of the post-MLP block LayerNorm $\mathbf{y}$ reads as

$$\mathbf{y} = (\mathcal{F}(\mathbf{x}\boldsymbol{\Gamma}\mathbf{E}) \odot \mathbf{x}\boldsymbol{\Gamma}\mathbf{B})\,\mathbf{G} + \mathbf{x}\boldsymbol{\Gamma}. \tag{6}$$

Similar principles as above together with basic properties of matrix norms yield

$$\|\mathbf{y}\| \propto \|\|\boldsymbol{\Gamma}\mathbf{E}\|\mathbf{x}\boldsymbol{\Gamma}\mathbf{B}\mathbf{G} + \mathbf{x}\boldsymbol{\Gamma}\|_F, \tag{7}$$

where we used the fact that $\|\mathbf{x}\| = 1$. Finally, the scaling factor computes as

$$\frac{\|\mathbf{y}\|}{\|\mathbf{x}\|} \propto \|\boldsymbol{\Gamma}(\|\boldsymbol{\Gamma}\mathbf{E}\|\mathbf{B}\mathbf{G} + \mathbf{I})\|_F. \tag{8}$$

### 3.5 SLaNC for the Attention Block

Next, we derive a formula for the scaling factor of the LayerNorm following the standard attention block with $h$ heads. As it can be seen in Fig. 2c, the most critical observation here is that the product of the $\mathrm{Softmax}$ output $\mathbf{S}^i$ of head $i$ with $\mathbf{V}^i$ results in a convex combination of the rows of the latter. The outputs $\{\mathbf{S}^i\mathbf{V}^i\}_{i=1}^h$ are concatenated, hence, the norm of the concatenated vector can be approximated by the norm of the concatenation of $\{\mathbf{x}\boldsymbol{\Gamma}\mathbf{W}_{\mathbf{V}}^i\}_{i=1}^h$ which is precisely $\mathbf{x}\boldsymbol{\Gamma}\mathbf{W}_{\mathbf{V}}$. We get

$$\|\mathbf{y}\| \propto \|\mathbf{x}\boldsymbol{\Gamma}\mathbf{W}_{\mathbf{V}}\mathbf{P} + \mathbf{x}\boldsymbol{\Gamma}\|_F = \|\mathbf{x}\boldsymbol{\Gamma}(\mathbf{W}_{\mathbf{V}}\mathbf{P} + \mathbf{I})\|_F, \tag{9}$$

and conclude that the following scale should be used in the post-attention LayerNorm operator

$$\frac{\|\mathbf{y}\|}{\|\mathbf{x}\|} \propto \|\boldsymbol{\Gamma}(\mathbf{W}_{\mathbf{V}}\mathbf{P} + \mathbf{I})\|_F. \tag{10}$$

---

[1]Since we are mainly focusing on the order of magnitude of the norms of the hidden states involved, without impact on accuracy we discard the additive biases $\boldsymbol{\beta}$ of the LayerNorm operator.

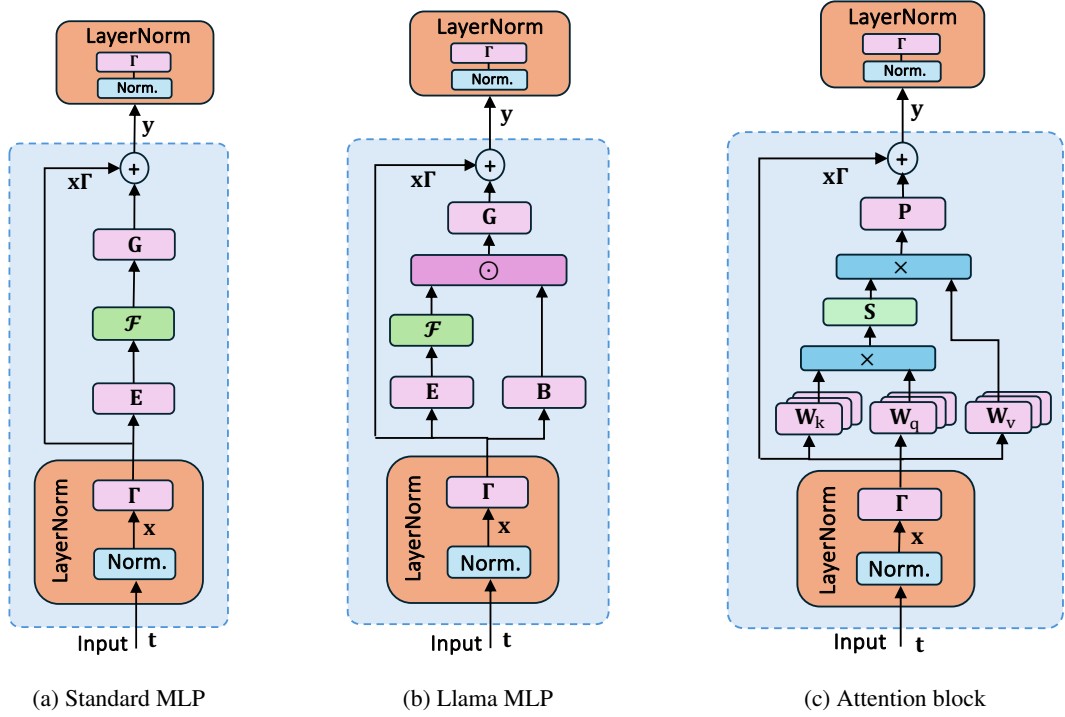

| (a) Standard MLP | (b) Llama MLP | (c) Attention block |

Figure 2: The compute flow between consecutive LayerNorms of various Transformers. Pink blocks with capital letters stand for linear layers with the corresponding weight matrices, green $\mathcal{F}$-blocks represent non-linearities and **S**-block represents $\mathrm{Softmax}$.

## 4 Experiments

To demonstrate the power of our SLaNC scaling technique, we present simulation results for Llama models. Note that the Llama architecture replaces the LayerNorm by Root Mean Squared Layer Normalization (RMSNorm) [36], which differs from the former only by omitting the mean $\mu$ subtraction in Eq. 2 and thus does not affect SLaNC scaling.

In our first experiment, we collected empirical statistics of the sums of squares in the denominators of the RMSNorm operators without scaling and with SLaNC scaling. To this end, we applied Llama-2-7b to Wikitext 2 dataset. Fig. 3c and 3a feature typical histograms in two consecutive RMSNorms of this model. For the lack of space, in this short article we illustrate the distributions in one specific Decoder 9, however we emphasize that these histograms do reflect the typical behavior of the data at hand. We see that in a significant number of cases, the sum of squares well exceeds the FP16 range and causes overflow. The SLaNC scaling changes the situation dramatically and not only shifts the histograms inside the FP16 range but also keeps safe margins on both edges of the range, as illustrated by Fig. 3d and 3b, respectively.

Next, we compared the perplexities of Llama models on the same Wikitext 2 dataset with the default FP32 implementation of RMSNorm and with the sum of squared accumulated in FP16 (all other operations from the default setup intact). Table 1 shows a significant degradation when the accumulation happens in FP16 exactly due to numerous overflows. This problem is completely resolved when the SLaNC scaling is applied. We also note that in all standard models, a small constant $\varepsilon$ is added to the variance of the input in the denominator of LayerNorm or RMSNorm operator. This way we can avoid division by zero in the case of underflow and improve the numerical stability. Since SLaNC scales are known ahead of time, we can easily apply them to the $\varepsilon$ constants as well (in fact, we divide $\varepsilon$ by the squared SLaNC scalings). As the bottom row of Table 1 demonstrates, now the FP16 SLaNC scaling can precisely reproduce the default FP32 values.

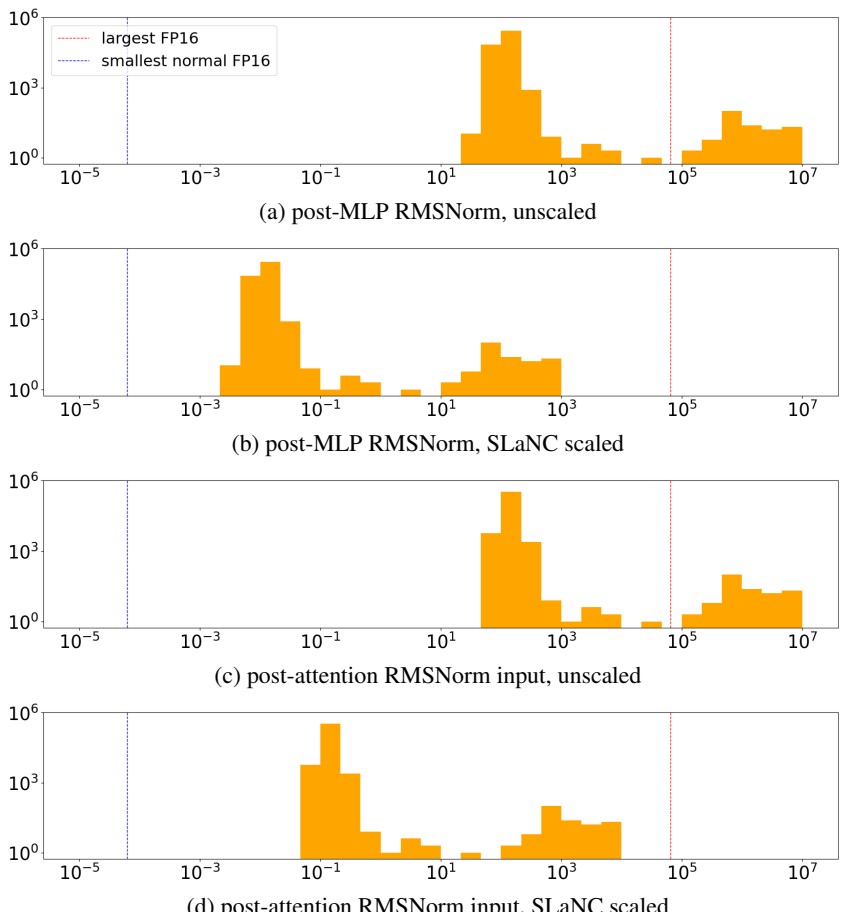

Figure 3: Empirical histograms of the sum of squares in RMSNorm layers of the 9th decoder in Llama-2-7b, calculated on wikitext2. The red vertical cut-off line sets the maximal representable FP16 value (65k) beyond which FP16 overflows, the blue line shows the minimal normal FP16 value. Histograms (b) and (d) show that after SLaNC scaling no overflow (or underflow) is detected and the RMSNorm is computed precisely. Note that these histograms built for the 9th decoder illustrate the typical behavior in all other layers too.

Table 1: Llama perplexity on Wikitext 2 with different RMSNorm computation modes.

| accumulation format | Llama-2-7b | Llama-2-13b | Llama-3-8b |
|---|---|---|---|
| FP32 | 5.116 | 4.574 | 5.538 |
| FP16 | 19.105 | 10.521 | 16.013 |
| FP16 + SLaNC | 5.116 | 4.573 | 5.539 |

## 5 Conclusion

In this paper, we present a novel SLaNC technique that makes LLM inference possible on FP16 accelerators without the need to cast LayerNorm operators into FP32. This theoretically grounded approach provides easy-to-use formulae for an offline computation of scaling factors for the inputs of LayerNorms. The SLaNC scaling factors enable precise computation of the LayerNorm in FP16 and provably avoid overflows and underflows. By keeping all the compute in FP16, the SLaNC algorithm enables low latency accurate compute, which is demonstrated by our extensive numerical simulations.

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
