# OpenReview forum: "SLaNC: Static LayerNorm Calibration"
_NeurIPS.cc/2024/Workshop/MLNCP — MLNCP Poster_

### Official Review · Reviewer_R8tx · 2024-10-05

**Rating:** 7
**Confidence:** 5

**Review:**

Paper content summary: This paper suggests a new way to compute Layernorm so that it can be done in FP16 without overflow or underflow. Due to the large magnitude of norm computation, the current standard is to upcast to FP32 before computing Layernorm. Therefore, the neat math the authors did provides a lossless transformation that helps compress the model distribution.

Reasons to accept:

- (see the paper summary)
- rigorous math deduction regarding norm approximation, linear algebra, and calculus. The notation and figures are also well documented.
- the reformulation of Layernorm provides a novel way to think about Layernorm and Transformer data flow
- the solution provided is comprehensive, covering standard transformer architecture and LLaMA models. The scaling works for both MLP and Attention blocks.
- the solution provided is lossless in terms of model accuracy, which provides a plug-and-play for a wide range of transformer models.
- the solution provided drastically reduces the magnitude of layernorm input to approximately 1.
- the solution generalizes well because it is statically determined by model weights, instead of depending on whether a calibration set is representative of the test set.

Here are a couple of questions to tighten the paper up:

- how much latency will be saved by using FP16 instead of FP32 for layernorm? I’m not completely convinced by the latency-saving motivation since most of the latency for LLM inference is matrix multiplication and data moving. Though the authors mention most accelerators support only FP16 for non-linear operations, in my personal industry experience, we have always used FP32 for Layernorm in NVIDIA GPUs. So I would like to see more evidence if the authors know of, on the impact this paper can bring.
- how much inference overhead will be added by doing additional scaling for Layernorm inputs?
- the conclusion claims that “*The SLaNC scaling factors* guarantee precise computation of the LayerNorm in FP16 and provably avoid overflows and underflow”. but I’m not sure if it guarantees no overflow/underflow or if it just drastically reduces the chance of overflow/underflow because it brings the norm of y close to 1. I’d like to hear the justification of the guarantee.

Open to raising the rating after the comments are addressed.

---

### Official Review · Reviewer_timy · 2024-10-06
**Promising offline LayerNorm quantization support**

**Rating:** 7
**Confidence:** 4

**Review:**

The overall judgment of the paper is positive, as the benefit of removing the calibration data outweighs the cons, which mainly concern the experimental section, which could be enlarged.

# Pros
- The work is clear, original enough within the workshop's scope, and significant for the topic.
- The techniques do not require modification of existing models, so they do not require retraining and seem to be deployable to any LLM.
- It provides a theoretical guarantee of avoiding underflow and overflow.

# Cons
- Efficiency is the main topic of the work. However, no instance of efficiency measurements are provided. What is the cost of the standard approach?
- In this regard, in Section 3.1, the sentence “This technique requires extra calibration data and significant computational overhead even for such a basic operation as LayerNorm, making this approach impractical. ” can be elaborated more. For instance, if the standard approach needs to be applied only once, it might not matter too much, whereas it becomes an issue if the models are updated frequently, but it really depends on the application.
- Please elaborate on the choice of “9th layer of LLaMA”, why the 9th?
- Showcasing results across all layers (for instance, mean) would be appreciated.
- The experimental section is lacking. Testing on other datasets and models is appreciated.

# Minor corrections:
- In Section 3.2 “liner layer” → “linear layer”

# Comments
- Could this technique also target stronger quantization strategies like int8 or lower?

---

### Decision · Program_Chairs · 2024-10-10

Accept (Poster)